# Trends in Nanoparticles for Leishmania Treatment: A Bibliometric and Network Analysis

**DOI:** 10.3390/diseases11040153

**Published:** 2023-10-28

**Authors:** Gabriel Mazón-Ortiz, Galo Cerda-Mejía, Eberto Gutiérrez Morales, Karel Diéguez-Santana, Juan M. Ruso, Humberto González-Díaz

**Affiliations:** 1Facultad Ciencias de la Vida, Facultad Ciencias de la Tierra y Agua, Universidad Regional Amazónica Ikiam, Parroquia Muyuna km 7 vía Alto Tena, Tena 150150, Napo, Ecuador; gabriel.mazon@ikiam.edu.ec (G.M.-O.); galo.cerda@ikiam.edu.ec (G.C.-M.); eberto.gutierrez1@ikiam.edu.ec (E.G.M.); 2Soft Matter and Molecular Biophysics Group, Department of Applied Physics and Institute of Materials (iMATUS), University of Santiago de Compostela, 15782 Santiago de Compostela, Spain; juanm.ruso@usc.es; 3Wood Engineering Department, University of Bio-Bio, Concepcion 4030000, Chile; 4Department of Organic and Inorganic Chemistry, University of the Basque Country UPV/EHU, 48940 Leioa, Spain; 5Basque Center for Biophysics CSIC-UPVEH, University of Basque Country UPV/EHU, 48940 Leioa, Spain; 6IKERBASQUE, Basque Foundation for Science, 48011 Bilbao, Spain

**Keywords:** nanomedicine, bibliometric analysis, network analysis, parasite diseases, leishmaniasis treatment, antiprotozoal agents

## Abstract

Leishmaniasis is a neglected tropical illness with a wide variety of clinical signs ranging from visceral to cutaneous symptoms, resulting in millions of new cases and thousands of fatalities reported annually. This article provides a bibliometric analysis of the main authors’ contributions, institutions, and nations in terms of productivity, citations, and bibliographic linkages to the application of nanoparticles (NPs) for the treatment of leishmania. The study is based on a sample of 524 Scopus documents from 1991 to 2022. Utilising the Bibliometrix R-Tool version 4.0 and VOSviewer software, version 1.6.17 the analysis was developed. We identified crucial subjects associated with the application of NPs in the field of antileishmanial development (NPs and drug formulation for leishmaniasis treatment, animal models, and experiments). We selected research topics that were out of date and oversaturated. Simultaneously, we proposed developing subjects based on multiple analyses of the corpus of published scientific literature (title, abstract, and keywords). Finally, the technique used contributed to the development of a broader and more specific “big picture” of nanomedicine research in antileishmanial studies for future projects.

## 1. Introduction

Leishmaniasis, an overlooked tropical illness, affects 12–15 million people globally, mostly in developing nations, due to subpar sanitation, insufficient protective measures (such as vector control), and inadequate sanitary infrastructures [1,2]. The most recent data from the WHO anticipates an annual occurrence of new cases ranging from 0.7 to 1 million [3], and 350 million people in 98 developing countries are at risk [1,4]. Leishmaniasis is the second most widespread pathogenic illness due to its high infection and morbidity rates [5].

According to de Souza, et al. [6], 21 of 53 Leishmania protozoan parasites cause this condition. Phlebotomine sandflies sucking contaminated blood spread it. This determines anthroponotic or zoonotic illness. Phlebotomine sandflies have exterior motile promastigotes and host macrophages have intracellular immotile amastigotes [7]. Amastigote to promastigote parasite transformation takes 4–25 days [8]. However, leishmaniasis has long been one of the most neglected tropical diseases [9]. Disease control requires chemotherapy. Antimonials and miltefosine, the main leishmaniasis treatments, have caused drug resistance in some areas, increasing the risk of (re)emergence and global expansion [9]. Due to a lack of diagnostic tools and effective vaccines, high disease prevalence requires safe and effective low-cost medications (chemotherapy is the first-line treatment for leishmaniasis, but it has side effects such as toxicity and chemoresistance) [10].

Consequently, new treatments are needed to control leishmaniasis. In this regard, the unique and easily tunable physicochemical properties of nanoparticles (NPs) can contribute to the detection of *Leishmania* spp. [11]. Nanocarriers for intracellular delivery of antileishmanial medications are being developed to improve therapeutic efficacy, reduce dosages, and reduce toxicity [1,6]. NPs have been successfully used to treat leishmania. Nanoparticles’ properties can help biomedicine cure parasite infections (large surface areas, adequate size, regular morphology, and electrical charge).

On the other hand, several reviews have focused on the application of nanotechnology in leishmaniasis treatment [1,10,12,13]. Moreno, Schwartz, Fernández, Sanmartín, Nguewa, Irache and Espuelas [13] examine the pros and cons of using NPs for topical treatment of cutaneous leishmaniasis (CL). The study by Akbari, Oryan and Hatam [12] reviews nano-technology methods for treating leishmaniasis, focusing on drug delivery to the target cell. Similarly, Saleem, Khursheed, Hano, Anjum and Anjum [1] examine the efficacy of several nanotechnology-based approaches and materials in treating leishmaniasis. Liposomes, metal, metal oxide, polymeric NPs, nanotubes, and nanovaccines were reviewed as potential anti-leishmanial disease therapeutic agents. Recently, Nafari, Cheraghipour, Sepahvand, Shahrokhi, Gabal and Mahmoudvand [10] studied the use of NPs as a therapy for treating various kinds of leishmaniasis. Their findings show that nanoparticles improve chemical drug quality, efficacy, and sustainability while lowering costs. However, none of the studies to date have examined the landscape of NPs for leishmaniasis treatment and research using bibliometrics and network analysis.

Bibliometric and scientific mapping techniques have been applied in several studies related to nanomedicine. Teles RHG, et al. [14] analysed the research papers published from 2012 to 2017 that explored the intersection of nanotechnology and triple-negative breast cancer. Ale Ebrahim S, et al. [15] examined “exosome” publication trends and cancer detection. Other bibliometric studies have also focused on parasitic diseases. For instance, Keighobadi, et al. [16] conducted a bibliometric analysis of global research in Scopus (1933–2019) on *Lophomonas* spp. Tantengco and Rojo [17] identified patterns in Southeast Asian schistosomiasis research articles (1908–2020). Ekici, et al. [18] analysed trends in the literature indexed in WoS over a 51-year period for the protozoan agent *Naegleria fowleri*. Ellis, et al. [19] reviewed parasitology’s main themes and research topics over the past 30 years. Specifically, regarding leishmaniasis treatment, Ramos, et al. [20] conducted a bibliometric analysis of Medline leishmaniasis research from 1945 to 2010. They observed a rise in the quantity of research papers within the realm of leishmaniasis. However, the use of nanoparticles was not considered.

The objective of this report is to conduct an exhaustive bibliometric analysis of papers published in the last 30 years on the use of NPs in leishmanial research. It lists the most influential authors, institutions, nations, documents, and bibliographic sources on the subject. The Bibliometrix R package and VOSviewer application are employed to depict the bibliometric data linked to the research visually. This involves the examination of keywords, citations, co-citations, and bibliographic matches. Additionally, emerging themes are highlighted through multiple correspondence analysis (MCA), and their distribution is visualized on thematic maps. Finally, this report seeks to define the most important NP themes and groups of themes in antileishmanial research and analyse their evolution over time.

The remainder of this work is organised as follows. The technique, including the research words and the flowchart outlining the bibliometric analysis procedure, is detailed in Section 2. Section 3 provides an overview of the citation systems utilised by prominent countries/regions, institutions, writers, and documents. Furthermore, in Section 4, the research trends in the utilisation of NPs in antileishmanial studies are investigated. Section 5 concludes the paper by recommending future research directions on this topic.

## 2. Material and Methods

This review employs bibliometric techniques to examine the literature concerning the use of NPs in antileishmanial applications. By conducting bibliometric analysis, we can assess specific aspects of academic research and academic journals, including journal citations, authors, and other metadata [21,22,23]. Citations can reveal links between an individual author and a particular topic, research approach, or collaborative work [24]. According to [21], academics can harness citation analysis to:(i)Explore the intellectual structures within research areas by initially creating a map of these structures.(ii)Assess the educational implications and sources of information.(iii)Track the dissemination of ideas and the flow of knowledge.(iv)Aid in information retrieval, organisation, and presentation.(v)Investigate the readership and usage patterns within the academic literature.

### Bibliometric Method

At first, bibliometric analysis focuses on pinpointing the foremost ten contributors based on authors, institutions or universities, and countries. It delves into the collaborative patterns among authors, institutions, and nations. Additionally, this approach consolidates the contributions within a research field, conducts a comprehensive literature review, dissects the principal topic or theme clusters within the domain, and puts forward potential avenues for future research. Our data source consisted of bibliographic records extracted from the Scopus database, encompassing peer-reviewed literature across a spectrum of fields, ranging from the sciences to the social sciences [25,26,27]. According to [28], bibliometric studies are objective literary content synthesisers. This paper uses bibliometrics to look back at 30 years of NPs for leishmaniasis treatment and research, from 1991 to 2022.

Three phases comprise the methodology used. Phase 1 consists of data collection, search methods, and data pre-processing. The study’s literature was compiled from the Scopus database, which includes abstracts and citations. Several searches with terms unique to NPs for leishmaniasis treatment and research (e.g., leishmania, leishmaniasis, visceral leishmania (VL), promastigote, amastigote, nanoparticle, leishmania species name, etc.) yielded a very small number of relevant papers. Therefore, the search phrase in Scopus was updated to TITLE-ABS-KEY (nanoparticle, OR nanoparticles, AND leishmania). Scopus bibliometric data for the requested keywords led to the choice of the years 1991 to 2022. The information was obtained on 1 January 2023. Articles for the year 2023 were excluded from the analysis because they still need to be completed. Once the total number of papers had been determined, the inclusion/exclusion criteria from Table 1 were applied to reduce the number of papers to a more precise total. This included irrelevant documents among the articles that were not related to what is being discussed or considered in the objective and scope of this work. In addition, filtering by subject area was carried out. All data was collected in CSV and BibTeX files and contained all complete records and cited references.

Phase 2 includes mapping, bibliometric analysis, and software development. Bibliometric analysis is a quantitative method for analysing academic literature that uses bibliographies to describe, appraise, and continue previously published research [29]. The methodology’s goal was to examine publications, citations, and information sources. Visualisation and bibliometric mapping of the papers consulted were carried out using the VOSviewer 1.6.17 programme and the Bibliometrix R package, version 4.0 [30]. Using bibliographic linkage, the identification of the most significant study clusters, researchers, and articles related to a specific research topic was conducted [31]. Next, a joint co-citation analysis was undertaken to categorise the work into numerous issues or topics based on its conceptual structure and to identify articles shared by publications.

Phase 3 consists of trend analysis in science. This phase involved an assessment of the current status and emerging patterns in knowledge, research groupings, forthcoming research topics, and key figures of reference that may help with research collaborations. Graphical representations were made using VOSviewer v1.6.17. This enabled us to chart the scientific landscape, encompassing research trends, authors, institutions, and publications [23,32]. Our experiments encompassed: (i) the creation of co-occurrence networks and overlap mapping of publication keywords and (ii) the establishment of co-occurrence networks and overlap mapping for publication titles and abstract texts. These co-occurrence analyses complemented the combined citation analysis conducted in Phase 2. Additionally, we conducted multiple correspondence analysis (MCA) and analyzed topic map distributions using the Bibliometrix R package. The comprehensive plan for our bibliometric investigation is illustrated in Figure 1.

## 3. Results

### 3.1. Data Analysis and Visualisation

A global survey of NPs in antileishmanial research yielded 576 articles from reputable journals and universities. These respected journals are scientific or scholarly journals with a solid reputation and are highly regarded in their respective fields of study. Once the inclusion/exclusion criteria from Table 1 were applied to concentrate the bibliometric analysis on NPs for leishmaniasis treatment, a total of 524 papers were recovered.

Based on the implementation of the TITLE-ABS-KEY search method, the Scopus dataset yielded a total of 524 unique documents on the chosen subject, with a total of 2255 authors and 5003 keywords. The 524 documents analysed comprised 254 bibliographic sources from journals, book chapters, conference papers, editorials, and reviews, of which 432 were articles. The remaining documents accounted for the remaining nine book chapters, 74 reviews, four conference papers, and five editorials. There was an average of 27.02 references per article. This finding suggests that a collection of papers has garnered a substantial number of citations.

In contrast, only a limited number of articles have received few citations. Furthermore, the collection included 13 articles by a single author on NPs for leishmaniasis treatment and research, as well as 510 documents co-authored by 2242 separate scholars. The average number of authors per publication was 6.54, indicating that most publications on this subject result from collaborative research. The proportion of international co-authorships was 29.77%, while the yearly growth rate was 13.3%. The statistical summaries for these data are shown in Table 2.

The distribution of nanomedicine application publications focusing on leishmanial research from 1991 to 2022 (1 January 2023) is displayed in Table 3. In the first twenty years, only 33 papers were published (1991–2010). However, from 2011 onwards, the number of publications on the subject increased. Eleven documents were published in 2011, 22 in 2012, 19 in 2013, 22 in 2014, and 29 in 2015. In the past seven years, productivity has increased, with 43 articles published in 2016, 34 in 2017, and 54 in 2018. In 2019, 57 papers were published, followed by 72 reports in 2020, 80 in 2021, and 48 in 2022. The results showed that academics are becoming more interested in this topic yearly.

Furthermore, Table 3 provides citation evaluation with seven levels, allowing a quality study of publications. The year 2017 had the most citations (1410), with an average of more than 41.47 citations/document and 6.91 citations/year. Even though the average number of citations for the 2001 article is much higher (1223 C/doc and 55.59 TC/TP-year), this work has been available to scholars for more than 20 years because it was one of the first in the field to be published.

Table 4 presents countries with seven or more publications in research into nanomedicine application in the antileishmanial field (identified by the nation of the corresponding author). Among the 14 countries, Iran tops the list, followed by Brazil, India, and Pakistan. These countries stand out in the research on nanoparticles for the treatment of leishmaniasis precisely because they are endemic countries for the disease and annually register cases of CL and VL, as the WHO figures show [33]. Figure 2 shows the ten countries with the highest number of reported cases of CL and VL in the year 2021, and these countries appear together with the Syrian Arab Republic and Afghanistan, among other tropical developing countries. As stated previously, Iran contributed the most to NPs for leishmaniasis treatment and research, with 98 research articles, accounting for 18.70% of all publications, followed by Brazil (96; 18.32%), India (80; 15.26%), Pakistan (53; 10.11%), and the United States (20; 3.82%). In the examination of the mean citation count per publication, Germany (14th in the number of publications) ranks first, with 218.29 citations/doc, followed by Italy (44) and the United States (41.10). According to [34], this metric is essential for comparing the citation strengths of various nations.

In the bibliometric analysis of the institutions, 1455 contributed papers on NPs for leishmaniasis treatment and research. Table 5 displays the statistical data for the top ten research institutes. Among these, four are based in Iran and Brazil, underscoring their dominant presence in this field. A correspondence between the most productive institutions and countries is shown, as these four countries occupy top positions in the country rankings.

Of the research institutions, the best-ranked field under analysis was Quaid-i-Azam University (52 papers, 9.92% of the total), which belongs to Pakistan. It is also the institution having the most significant number of citations (1877), h-index (26), and average citations per document (36.1), which indicates that research of this institution has had the greatest impact in the field of nanomedicine application in anti-leishmanial activity.

### 3.2. Distribution of Bibliographic Sources

A total of 524 documents were published in 254 scholarly sources. Table 6 presents the ten prolific journals, which account for 126 publications, 24.05% of all research papers. The International Journal of Nanomedicine is the most productive, with 21 (4.01%) research documents, 1163 citations, and a score of 15 on the h-index. This journal, publishing original research from the areas of pharmaceutical and material sciences, is an important research platform focusing on pharmacology, toxicology, pharmaceutics, nanoscience, and nanotechnology, and specifically in the applications of nanotechnology in the biomedical field, as work related to the development of possible clinical applications of nanoparticles in the diagnosis, prevention, and treatment of diseases. Nanomedicine is in second place in terms of number of publications, total citations, and h-index with 19 documents, 546 citations, and a score of 13 on the h-index, respectively. This journal specialises in medicine, bioengineering, and materials science, more specifically, in nanoscience, nanotechnology, biotechnology, and applied microbiology. It includes original papers related to developing new nanotechnological therapeutic approaches for diagnosing, preventing, and treating diseases and conditions, which are closely related to the object of this bibliometric study. Acta Tropica ranks third (16; 3.05%); this journal specialises in topics relevant to human and animal health in the tropics and subtropics. It addresses the areas of immunology, microbiology, and tropical medicine, more specifically, parasitology and infectious diseases, which include relevant aspects of clinical diseases and the treatment of parasites such as leishmania.

Table 7 lists the ten most cited writers in terms of total publications (TP), total citations (TC), and citations per publication (C/P), taking into consideration the high degree of collaboration. The author with the most publications was Anuradha Dube from India (16 documents), followed by Khamesipour, Ali and Rafati, Sima, from Iran (with 15 and 14 articles, respectively). However, in terms of C/P value, Bağırova, Melahat from Azerbaijan has the greatest value, with 59.1, followed by Abamor, Emrah Şefik (Turkey, 50 C/P), and Shinwari, Zabta Khan (Pakistan, 47.2 C/P), respectively. The authors above are widely recognised authorities in utilising NPs as anti-protozoal agents, having published groundbreaking research that substantially contributes to the advancement of theoretical knowledge and practical applications in this area of study. The indices (h, g, and m) were provided as indicators of citation and productivity, respectively. Dube, Anuradha and Rafati, Sima present the highest values of the h-index (14 and 12) and g-index (16 and 14). In the case of the m-index, Abbasi, Banzeer Ahsan and Iqbal, Javed show a value of 1.6.

Relevant publications: Table 8 lists the ten most cited publications on NPs for leishmaniasis treatment and research.

The first article, “Nanosuspensions as particulate drug formulations in therapy: Rationale for development and what we can expect for the future” [35], shows the highest citations per year (53.17) and 1223 citations. This work was published in 2001, one of the first ten years considered in this study. This work addressed the production of drug NPs (nanosuspensions) to solve drug solubility difficulties. Based on lab-produced NPs, it describes their general therapeutic application. It demonstrates surface-modified drug NPs for delivery to the brain as well as mucoadhesive nanosuspensions for oral administration. The second paper, “The potential of nitric oxide releasing therapies as antimicrobial agents”, focuses on analysing different delivery systems of nitric oxide (NO) to inhibit antimicrobial activity. Specifically, it describes the use of NPs as delivery systems for NO molecules, e.g., diazeniumdiolate-coated NPs that are effective against pathogens. Additionally, it describes using molecules containing NO-bound particles (acidified nitrite, S-nitrosothiols) to treat leishmaniasis infections [36]. The third most cited article, “Quantum dots in imaging, drug delivery and sensor applications”, has 326 citations and ranks second in terms of citations per year (46.57). This study examines medical applications of quantum dots (QDs) (nanoscale semiconductor crystals). In the case of leishmaniasis, they demonstrate the use of magnetic beads and CdSe QDs as sensors for the detection of leishmania-specific surface antigens [37].

The fourth article is entitled “Antimicrobial effects of TiO_2_ and Ag_2_O nanoparticles against drug-resistant bacteria and leishmania parasites” [38] (279 citations). It reviews the distinctive features of NPs and their mechanism of action as antibacterial and antileishmanial agents of metal oxide NPs, particularly TiO_2_ and Ag_2_O produced by visible and UV light against drug-resistant species. Another work, from 2011, was the fifth more cited document (245 citations), headlined “Recent advances in leishmaniasis treatment” [39]. This article provides an overview of the available medical treatments for Leishmania infections, including breakthroughs in research on plant and synthetic substances as potential drugs to treat the disease. It emphasises technologies based on NPs, liposomes, cochleates, and non-specific lipid transfer proteins while discussing particular drug delivery systems for the creation of new chemotherapeutics. The sixth article (181 citations) is entitled “Formulation of amphotericin B as nanosuspension for oral administration” [40]. The authors designed a novel oral drug delivery system for the experimental treatment of VL. A nanosuspension of amphotericin B cut the number of parasites in the livers of mice by 28.6% compared to mice that weren’t treated.

Another paper from 2011 and the Allahverdiyev et al. group was “Antileishmanial effect of silver nanoparticles and their enhanced antiparasitic activity under ultraviolet light” [41]. It was the seventh document in terms of citations (164). The authors report the effects of Ag NPs on parasites of *Leishmania tropica*. Biological parameters such as morphology, metabolic activity, proliferation, infectivity, and survival in host cells in vitro were analysed in the parasite species. In the presence of UV light, Ag NPs inhibited the proliferation and metabolic activity of promastigotes and the survival of amastigotes in host cells. The eighth article, “PLGA nanoparticles and nanosuspensions containing amphotericin B: Potent in vitro and in vivo alternatives to Fungizone and AmBisome”, was published in 2012 and has been cited 124 times in the Scopus database, with an annual citation rate of 10.33%. The nanomedicines developed in this study (poly(D,L-lactide-co-glycolide) nanoparticles and amphotericin B nanosuspension) were as effective as the free drug against *Leishmania infantum* promastigotes and intracellular amastigotes [42]. The ninth most cited paper (118 citations) is “*Sageretia thea* (*Osbeck*.) modulated biosynthesis of NiO nanoparticles and their in vitro pharmacognostic, antioxidant, and cytotoxic potential” [43]. This paper studies the effects of UV-exposed NPs on six pathogenic bacterial strains. The cytotoxicity of the NPs against *Leishmania tropica* KWH23 promastigotes and amastigotes was also measured. Finally, the tenth document (117 citations), “Recent updates and perspectives on leishmaniasis”, was published in the Journal of Infection in Developing Countries. In this review, Savoia [44] discusses diagnostic, chemotherapeutic, and immunisation strategies utilised to control leishmaniasis, including using NPs.

### 3.3. Bibliometric Coupling of the Connections among Documents, Authors, and Journals

Initially, a network was constructed to explore collaborative research and citation relationships between authors and journals through coupling and co-citation. Bibliographic coupling refers explicitly to the relationship between articles. In Figure 3, you can observe the bibliographic coupling of articles, along with visual cues denoting clusters through different colors. The size of the nodes in the figure corresponds to the total number of citations received by each article. Furthermore, the distance or proximity between publications in a network reveals the bibliographic connectivity between nodes/documents. For example, the proximity of the two articles suggests that they share a substantial number of references [31]. This study only included papers with 15 or more citations. Two hundred sixty connected articles out of 524 publications met the minimum requirement. The VOSviewer program was used to compute the total link strength (TLS) and the number of citations for each publication, and then select the papers with the most substantial TLS values. The paper by Müller, Jacobs and Kayser [35] was the most cited publication (1223 citations), but only three TLS values are depicted in Figure 2. However, the strongest article was by Bruni, et al. [45], with 444 TLS and 57 citations.

Initial bibliometric author linking was performed to identify author collaboration in NPs for leishmaniasis research and treatment (as shown in Figure 4). In terms of bibliographic links, seven clusters are presented, and the top four authors according to TLS are Abassi, B.A (8925), Iqbal, J. (8925), Kanwal, S. (7777), and Mahmood, T (7777). The first two appear in the green cluster and are active collaborators with other leading researchers such as Shinwari, S.K. (6883), Khalil, A.T (6565), and Maaza, M. (6226). Another relevant author is Khamesipour, A (7559) (cluster blue), who is accompanied by Rafati, S. (6484), Jaafari, M.R. (5325), and Zahedifard, F. (4126). In cluster red, Dube, A (4003), Nadman, A. (3858), Shahnaz, G. (3791), and Abamor, E.S. (2935) stand out.

Bibliographic linking sources: Figure 5 illustrates the bibliographic linkage between the primary sources. There are three clusters, and some of the most influential publications stand out. The green cluster is, for example, dominated by Nanomedicine, which contains 19 articles and a TLS of 1522, followed by the International Journal of Nanomedicine, (21 documents and 1123 TLS) (see green cluster). In the third place, Acta tropica (with 16 docs, 988 TLS), is connected in the red cluster with Experimental Parasitology (10 documents and 508 TLS). The International Journal of Pharmaceutics is the fourth bibliographic source in terms of TLS, with 10 docs, and a TLS score of 982 (see blue cluster).

Co-citation network analysis: In the VOSviewer programme, a co-citation network was built for the key authors. Research communities were formed through author interactions relying on co-citations. Author citations were required to total at least 25, and 463 of the 50,987 authors met this condition. Figure 6 depicts four groups with citation bubbles; a larger bubble indicates a more credible source. The most prominent authors are highlighted in the various color-coded groups, including Sundar, S. (521 citations, 54,938 TLS), Khamesipour, A. (314, 49,675 TLS), and Dube, A. (272, 41,831 TLS). In addition to Sundar, S, commonly referenced authors in the red group include Croft, S. L. (346 citations, 28,449 TLS) and Alvar, J. (187 references, 16,448 TLS).

### 3.4. Scientific Landscapes/Trend Analysis

Based on three significant information aspects of the data corpus, a scientific landscape analysis was undertaken for NPs for leishmaniasis treatment and research (title, abstract, and keywords). This study employed three main methods to build things: MCA, thematic map distribution, and co-occurrence network analysis. The co-occurrence network analysis involved creating a keyword co-occurrence graph to deduce prevalent themes and subjects from broader phrases. The size of the circles in the graph corresponds to the frequency of key phrases found in articles, while their proximity reflects the degree of co-occurrence between them. This network was constructed using co-occurrence analysis and the complete count method. With a maximum frequency of 25, only 145 of the 5709 keywords discovered in the search of all keywords are displayed (see Figure 7). The co-occurrence network is composed of four topic clusters interacting via 9892 links, with a total link count of 155,548. The following describes each cluster:

Green cluster: NPs for antiprotozoal treatment: This group comprises a total of 48 keywords associated with using NPs for leishmaniasis. The primary applications highlighted as study topics are as follows: nanoparticle 276, 7340 TLS; nanoparticles 273, 6662 TLS; leishmaniasis 209, 5172 TLS; antiprotozoal agents 162, 4737 TLS; drug delivery system 168, 4665 TLS; and antiprotozoal agent 147, 4380 TLS.

Red cluster: Controlled studies on leishmanial fields: this cluster contains 53 elements focusing on studies for the treatment of leishmanial diseases, mainly under controlled conditions such as in vitro studies. The principal research applications highlighted are nonhuman, with 357 occurrences (9224 TLS), controlled study (253 occurrences, 7002 TLS), chemistry (187 occurrences, 5226 TLS), particle size (172 occurrences, 4756 TLS), and in vitro study (151 occurrences, 4501 TLS).

Blue cluster: Animal experiment and model for leishmanial study. This cluster incorporated 354 items focused on studies involving animals cell lines for the treatment of parasitic diseases such as leishmania. The main terms in the cluster are animals (253 occurrence, 7233 TLS), animal (6276 TLS), mouse (6318 TLS), animal experiment (4724 TLS), and mice (4607 TLS). Despite the unique characteristics of each cluster, a highly interconnected network is created.

Only 69 of the 1208 keywords identified by the authors’ keyword analysis are presented, with a maximum frequency of five (see Figure 8). The total number of links in the graph is 529 and TLS is 1093. NP topics are frequently observed in the field of antileishmanial research. Cluster 1, which contains the topic of drugs for VL, is represented in red. This cluster has 18 elements, including amastigotes, amphotericin B, curcumin, gold nanoparticle, *Leishmania major*, *Leishmania donovani*, *Leishmania infantum*, miltefosine, pentamidine, PLGA, and visceral leishmaniasis. The green cluster is focused on the use of nanotechnology in leishmaniasis treatment. It includes 15 terms related to drug delivery, solid lipid NPs, vaccines, and leishmaniasis treatment. The blue cluster stands out for the topic of green synthesis and biological activity and includes 10 terms, including various biological activities such as antimicrobial, antibacterial, antileishmanial, anticancer, and antioxidant.

In Cluster 4, ten items, which are displayed in gold, deal with NP drug delivery systems for leishmaniasis chemotherapy, specifically *Leishmania amazonensis*, and include liposomes, PLGA, antileishmanial activity, chemotherapy, drug delivery, toxicity, and drug delivery systems. Cluster 5, in purple, groups nine terms and shows macrophage topics in anti-leishmanial activity and includes anti-leishmanial activity, vaccine, extracellular vesicles, cytokines, macrophage, and PLGA nanoparticles. Finally, the light blue cluster stands out for the terms nanomedicine and macrophage targeting. It includes seven terms: glucantime, macrophage targeting, chitosan, mannose receptors, Leishmania major, meglumine antimoniate, and nanomedicine.

According to TLS, the top ten author keywords are: leishmaniasis (187), nanoparticles (182), leishmania (132), amphotericin B (108), visceral leishmaniasis (96), nanotechnology (85), cutaneous leishmaniasis (66), drug delivery (63), antileishmanial (59), and liposomes (54).

## 4. Discussion

### 4.1. Conceptual Structure Map

Utilising bibliographic clustering, antileishmanial research on the utilisation of nanomedicine in the field of parasitic diseases is initially organised. Similar subject matters are clustered together. In addition, the density of the clusters can be utilised as a measure of the breadth of research conducted in the particular subject area. Dense clusters denote thoroughly researched themes, whereas sparse clusters provide room for additional research. Multiple correspondence analysis of the conceptual structure map for the purpose of examining bibliographic clusters is depicted in Figure 8. Initially, the topics were divided into the two groups displayed below.

Cluster 1: NPs and drug formulation for leishmaniasis treatment.

The cluster with the highest density is situated inside the red region depicted in Figure 8. The aforementioned collective encompasses the overwhelming majority of acknowledged and published recent scholarly investigations. This cluster exhibits a high degree of dominance. In terms of study topics, bulk articles in this field focus on drug formulation, nanoparticles, drug delivery systems, visceral leishmaniasis, particle size, amphotericin B (which is a widely used drug for the treatment of the disease), chemistry, and diverse leishmania species names. Consequently, this cluster contains a lot of eminent researchers and substantial contributions. Additional terms include human, nonhuman, animal, antiprotozoal agents, drug efficacy, and parasitology.

Cluster 2: Animal models and experiments. This cluster appears blue in Figure 9. It only includes twelve terms: mice, promastigote, controlled study, zeta potential, in vitro study, mouse, animal cell, animal model, female, gag albino mouse, mice inbred balb c, and animal experiment. As can be seen, they are all related to animal testing, mainly on mice. Using animals in research is a common way to find and develop new treatments for leishmania. It is also an important step in the field of research into how NPs can be used to treat disease.

### 4.2. Strategic Diagram of Research Trends in Publications

We examined research trends using approach diagrams (Figure 10) to visually represent the key research areas within the NPs in antileishmanial field publications. This analysis involved examining the distribution of thematic maps, with a specific focus on the author-generated keywords found in the most influential articles in the corpus, and utilised circles to categorise them into four research areas. The density of circles serves as a metric for gauging the level of subject development and centrality, which is directly proportional to the significance of the issue within the broader study domain. The value and standing of it are connected to factors such as the quantity of publications, primary research citations, the number of researchers involved, and so forth. The “driving themes” on the strategy diagram have been thoroughly developed internally and are tightly linked to other research areas. The “highly developed and isolated notions” (upper left quadrant) are more internally developed but have less scientific value. “Developing or declining themes”, located in the lower left quadrant, represent developing concepts that do not have research relevance and may not attain research significance in the future.

On the contrary, even though they experienced internal growth, themes that are declining become obsolete due to current theoretical developments. Lastly, “fundamental and cross-cutting themes”, placed in the lower right quadrant, indicate relevance to modern research topics despite a lack of internal development [46]. In our sample size, applications of nanotechnology in leishmaniasis are driving themes. Photodynamic therapy for leishmaniasis treatment is one of the most highly developed and isolated topics. According to [47], photodynamic therapy with a photosensitive or photosensitising substance capable of creating reactive oxygen species to which Leishmania parasites are sensitive has emerged as an alternate treatment for CL. Another niche theme is related to gold nanoparticles and pentamidine against *Leishmania infantum* and saponin-loaded PLGA NPs for intracellular drug delivery in Leishmania-infected macrophages. Macrophages and cytokines in anti-leishmanial activity are a declining/emerging topic. In addition, the investigation of the cytotoxic effects of silver nanoparticles (NPs) on Leishmania, the use of chitosan NPs for the treatment of CL, and the development of amphotericin B-loaded NPs for the treatment of VL emerge as pivotal and multidisciplinary research topics.

Next, we contribute to the scholarly discourse by spotlighting the evolving quantitative research trends in the field of antileishmanial studies. Figure 11 illustrates the annual patterns of research activities. Broadly, these publication trends can be categorized into two distinct phases. The initial phase of research centers on the years spanning from 1994 to 2011. Most of the papers in the first 20 years examined the mechanism of H_2_O_2_ toxicity to promastigotes and the use of primaquine-loaded poly(D,L-lactide) NPs to treat Leishmania species. The second category of literature encompasses works published from 2012 to the present. The focus on nanomedicine development is gaining ground, mainly with regards to different types of NPs that are used to transport drugs or inhibit antileishmanial activity. The most frequently occurring terms were non-human, nanoparticles, animals, amphotericin B, *Leishmania donovani*, visceral leishmaniasis, and drug effect.

The high surface energy or geometric energy of NPs allows them to have many applications in the treatment of diseases. As proposed by [48], infection by viruses can be prevented by controlling the fractal dimension of the virus surface using nanotechnology, which can also be applied to treat leishmaniasis. The application of NPs with drugs acts on the parasites. Moreover, the size of the NP is important for cellular uptake, its interactions with the immune system, and its elimination from the body [49]. For example, in VL, the parasite mainly infects macrophages in the liver and spleen. Therefore, uptake studies of macrophage cell lines using different NPs could provide a strategy to improve drug delivery for VL [50]. Thus, geometric potential theory and fractal dimensions related to the size and distribution of nanoparticles can be considered in the treatment of leishmania. Another aspect to consider could be thermal therapy. In that sense, it has been proven by [51] that thermal therapy enhances the metabolic activity of the infected area to cope with pathogens and viruses. In the case of leishmania, treatment with NP has not been addressed, but could be another topic area for future reviews.

Furthermore, machine learning (ML) techniques have been widely used in nanomedicine to aid in the discovery of dual antibacterial drug-nanoparticle systems [52,53], or to predict the biological fate and properties of a variety of NPs relevant to their biomedical applications [54]. Furthermore, Singh et al. [55] argued that these tools can improve the simulation and modelling processes for nanotoxicology and nanotherapeutics and support the development of safe nanomedicinal products. Meanwhile, Goonoo et al. [56] posited that ML techniques can be employed to characterise leishmaniasis wounds, monitor wound healing, monitor the success of intervention strategies for wound healing, and address the lack of expertise in low-resource settings. In this way, studies of nanotechnology in biomedical research and medicine will continue to grow and help lead to the creation of possible materials that could be used to treat diseases such as Leishmania.

## 5. Conclusions

This study conducted a bibliometric analysis to examine the body of literature related to the utilization of nanoparticles (NPs) in leishmaniasis research. A comprehensive evaluation of pertinent literature spanning from 1991 to 2022 was carried out, involving the retrieval of references, keywords, and author information from a total of 473 Scopus database entries. The exponential growth of studies centered on the application of NPs in leishmaniasis research from 2006 to 2022 underscores the substantial interest of the scientific community in this area. The bibliometric analysis and exploration of scientific landscapes and trends were performed using the Bibliometrix R package and the VOSviewer application.

In terms of NPs application in leishmaniasis papers, Iran supplied the most research to the leishmaniasis study, followed by Brazil, India, Pakistan, and the United States. Iran accounts for 18.70% (98 out of 524) of global scientific output. India has the greatest number of citations (2110). As a research institute, Quaid-i-Azam University (Pakistan) published the most literature (52 papers, 9.92% of total), followed by Tehran University of Medical Sciences (Iran), with 31 documents. Furthermore, Iran and Brazil have four of the top ten research institutes.

Additionally, our analysis revealed that the top ten academic sites were responsible for 24% of the overall contribution to the existing body of literature on NPs in leishmaniasis, with the International Journal of Nanomedicine is identified as the most prolific source, with Nanomedicine and Acta Tropica following suit. The most productive researchers are Dube, Anuradha, Rafati, Sima, and Khamesipour, Ali. Müller, R.H.’s “Nanosuspensions as particulate drug formulations in therapy: Rationale for development and what we can expect in the future”, which was published in Adv Drug Deliv Rev in 2001, has been cited the most (1223 times).

The data presented in this article are dynamic and will evolve over time, as research on the application of NPs in sectors related to parasitic disorders develops each year rapidly. Therefore, it is recommended that this study be repeated in the following years. In addition, data came from the Scopus database. Consequently, the limitations of the database may have had an impact on the outcomes of this inquiry. Bibliographic mapping was performed with VOSviewer to look into publication structures, citations, and research trends. Co-citations, bibliographic coupling, and author keyword co-occurrence were all examined. Current research on the use of NPs in the leishmaniasis field focuses on NPs and drug formulation for leishmaniasis treatment, animal models, and experiments in developing new treatments for leishmania. The co-occurrence network analysis reveals the presence of four theme clusters connected through 9892 links, with a TLS of 155,548. The main themes that are highly interconnected are NPs for antiprotozoal treatment, controlled studies on leishmanial fields, and animal experiments and models for leishmanial study.

In the coming years, it is expected that research in the antileishmanial field will maintain its emphasis on the study of NP research; principally, metal and metal oxide NPs have proven to be effective in inhibiting resistant biological strains. Nevertheless, cytotoxicity remains a difficult challenge for their development and application, so strategies are needed to decrease direct risks to human health. In that sense, green synthesis can contribute to avoiding the production of unwanted or harmful by-products and be more sustainable and environmentally and human-friendly. On the other hand, coencapsulation systems of drugs in nanocarriers should be considered in order to optimise the leishmanicidal effect and avoid resistance of *Leishmania* parasites. Additionally, using techniques from artificial intelligence can help predict nanotransporter systems that are effective against leishmaniasis. The most important NPs in leishmaniasis research have undergone significant changes throughout the examined period. Initially, a significant portion of scholarly literature focused on investigating the underlying mechanism of H_2_O_2_ toxicity to promastigotes and the use of primaquine-loaded poly(D,L-lactide) NPs for treatment of Leishmania species (years 1994 to 2011). Now, the focus on nanomedicine development is gaining ground, mainly with regards to different types of NPs that are used to transport drugs or inhibit antileishmanial activity (years 2012 to 2022).

Lastly, the scientometrics analysis offers a comprehensive and detailed overview of the present status of global nanomedicine research in the field of antileishmanial studies. It also provides valuable insights that can assist in shaping research projects and innovations in the countries engaged in this research. Furthermore, it aids in identifying potential collaborators and expanding research endeavors. Even though the article’s conclusions don’t cover everything, academics interested in how NPs can be used to treat parasitic diseases will find this work very useful and instructive.

## Figures and Tables

**Figure 1 diseases-11-00153-f001:**
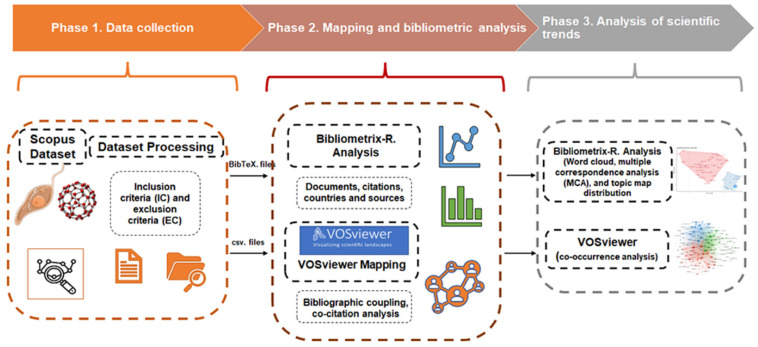
Bibliometric analysis workflow.

**Figure 2 diseases-11-00153-f002:**
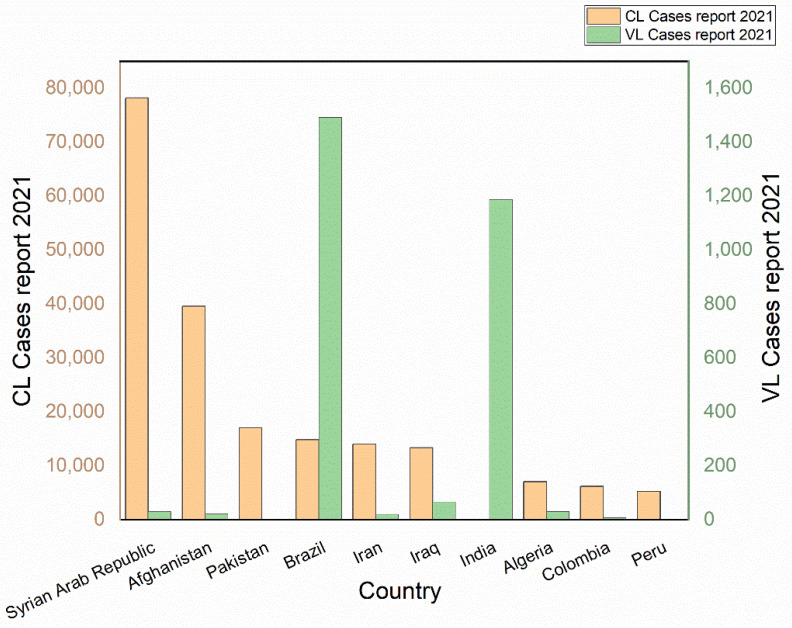
Top ten countries with the highest number of reported cases of CL and VL.

**Figure 3 diseases-11-00153-f003:**
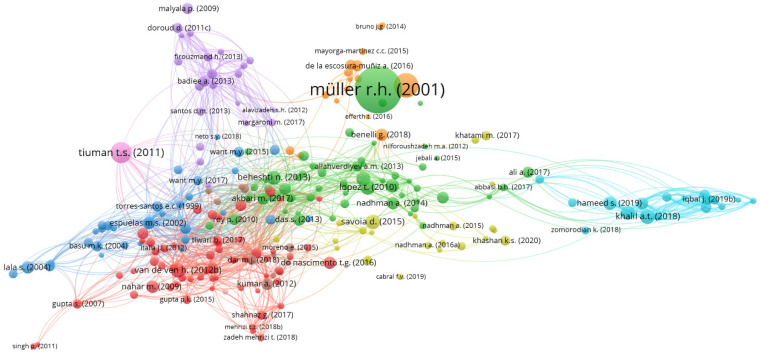
Bibliographic linking of research papers.

**Figure 4 diseases-11-00153-f004:**
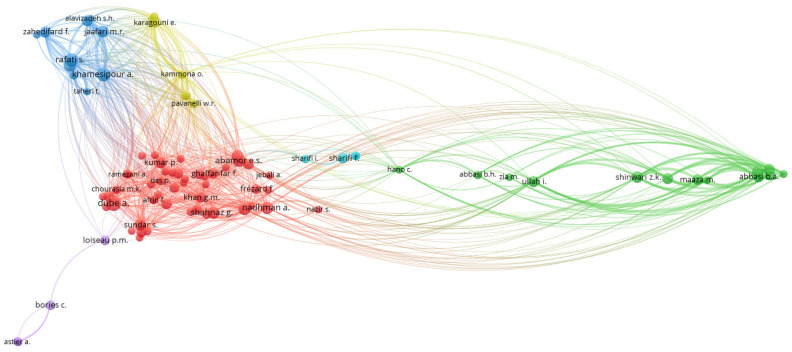
Bibliographic linkage of authors.

**Figure 5 diseases-11-00153-f005:**
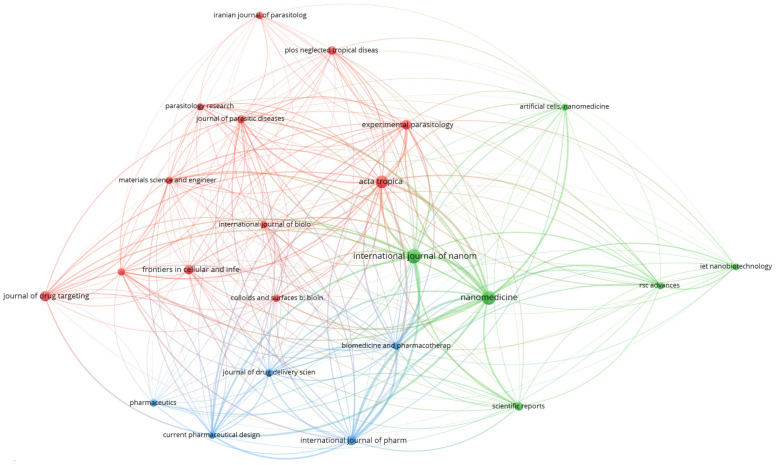
Bibliographic referencing of the primary sources.

**Figure 6 diseases-11-00153-f006:**
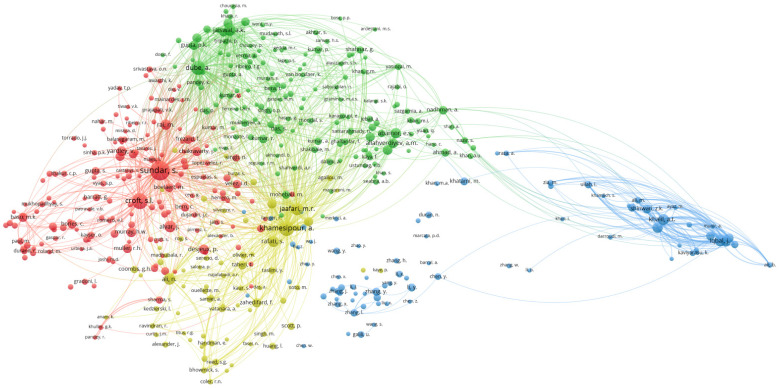
Co-citation network.

**Figure 7 diseases-11-00153-f007:**
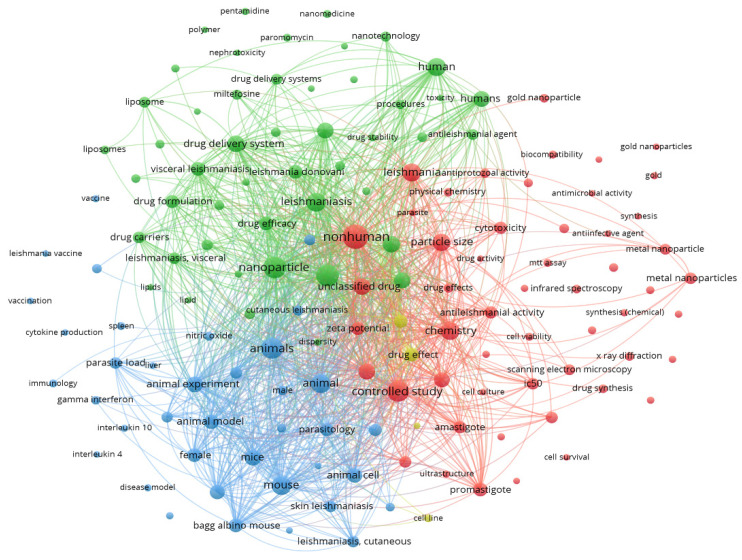
Keyword co-occurrences for overall keywords.

**Figure 8 diseases-11-00153-f008:**
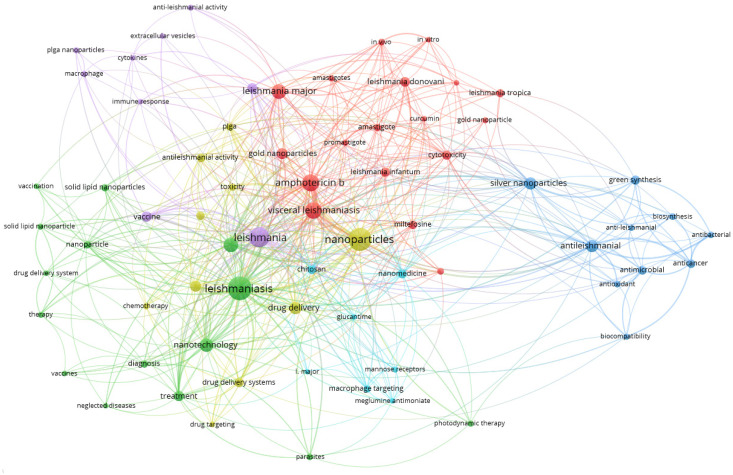
Co-occurrence of authors’ keywords.

**Figure 9 diseases-11-00153-f009:**
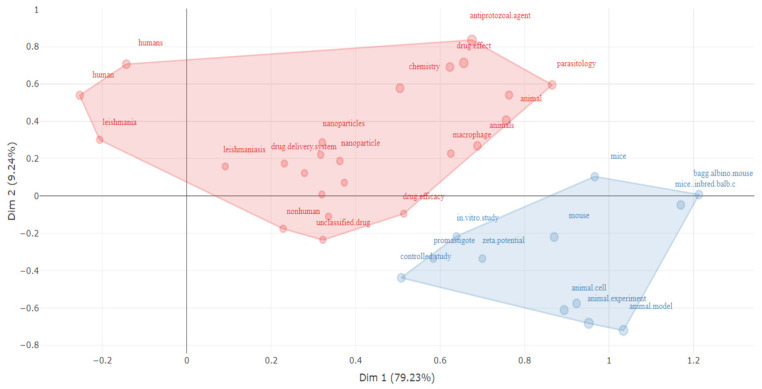
Diagram of the conceptual structure (k = 2 topic groups).

**Figure 10 diseases-11-00153-f010:**
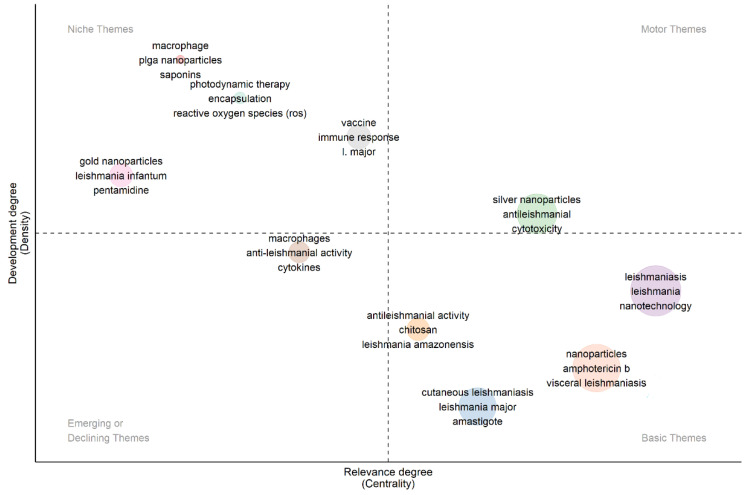
Diagram illustrating the research trends regarding the use of NP in leishmaniasis research.

**Figure 11 diseases-11-00153-f011:**
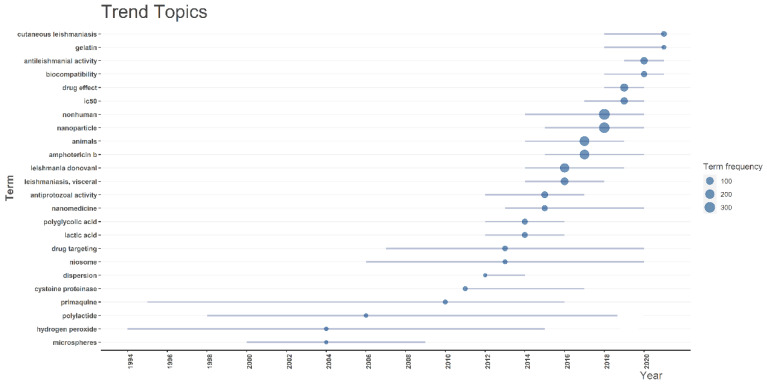
Trend topics keyword plus.

**Table 1 diseases-11-00153-t001:** Criteria for including and excluding the dataset from retrieval.

	Code	Criteria
Inclusion criteria (IC)	IC 1	Articles containing one of the keywords in either title, abstract, or keywords.
	IC 2	These documents are published in English
	IC 3	Publication date
	IC 4	Journal articles, conference papers, and book chapters
Exclusion criteria (EC)	EC 1	Articles with publication stage “in press”.
	EC 2	Irrelevant and incomplete documents
	EC 3	Unrelated sub-themes documents

**Table 2 diseases-11-00153-t002:** Summary of bibliographic statistics on the use of NPs for leishmaniasis treatment and research.

Description	Results
Timespan	1991–2022
Sources (journals, books, etc.)	254
Documents	524
Average years from publication	5.95
Average citations per document	27.02
Annual growth rate, %	13.3
References	30,188
Document types	
Article	432
Book chapter	9
Conference paper	4
Editorial	5
Review	74
Document contents	
Keywords plus (ID)	5003
Author’s keywords (DE)	1206
Authors	
Authors	2255
Authors of single-authored documents	13
Authors collaboration	
Single-authored docs	14
Co-authors per doc	6.54
International co-authorships, %	29.77

**Table 3 diseases-11-00153-t003:** Annual evolution of citations of the application of NPs for leishmaniasis treatment and research.

Year	TP	TC	TC/TP	(TC/TP)-Year	Citable Years
1991	1	53	53.00	1.66	32
1992	2	132	66.00	2.13	31
1994	2	88	44.00	1.52	29
1995	2	97	48.50	1.73	28
1997	3	110	36.67	1.41	26
1998	1	45	45.00	1.80	25
1999	1	62	62.00	2.58	24
2000	2	71	35.50	1.54	23
2001	1	1,223	1,223.00	55.59	22
2002	2	129	64.50	3.07	21
2003	1	181	181.00	9.05	20
2004	3	193	64.33	3.39	19
2005	1	63	63.00	3.50	18
2006	1	30	30.00	1.76	17
2007	1	48	48.00	3.00	16
2008	3	132	44.00	2.93	15
2009	3	171	57.00	4.07	14
2010	3	227	75.67	5.82	13
2011	11	1077	97.91	8.16	12
2012	22	918	41.73	3.79	11
2013	19	921	48.47	4.85	10
2014	22	640	29.09	3.23	9
2015	29	1069	36.86	4.61	8
2016	43	1013	23.56	3.37	7
2017	34	1410	41.47	6.91	6
2018	54	1353	25.06	5.01	5
2019	57	1113	19.53	4.88	4
2020	72	1009	14.01	4.67	3
2021	80	493	6.16	3.08	2
2022	48	86	1.79	1.79	1

Note: TP: total publications, TC: total citation frequency.

**Table 4 diseases-11-00153-t004:** Countries with seven or more publications in research on the application of nanomedicine in the antileishmanial field.

Country	TP	Contribution Rate (%)	TC	Average Citations per Document
Iran	98	18.70%	1802	18.39
Brazil	96	18.32%	1754	18.27
India	80	15.26%	2110	26.38
Pakistan	53	10.11%	1654	31.21
United States	20	3.82%	822	41.10
Spain	17	3.24%	393	23.12
France	15	2.86%	452	30.13
Turkey	12	2.29%	235	19.58
Iraq	11	2.10%	105	9.55
Italy	10	1.91%	440	44.00
Saudi Arabia	9	1.72%	186	20.67
Greece	8	1.53%	184	23.00
Portugal	7	1.34%	120	17.14
Germany	7	1.34%	1528	218.29

Abbreviations available in Table 3.

**Table 5 diseases-11-00153-t005:** The ten most prolific academic institutions in NPs for leishmaniasis treatment and research.

No.	Institute	TP	Contribution Rate (%)	TC	h-Index	Average Citations per Document	Country
1	Quaid-i-Azam University	52	9.92%	1877	26	36.1	Pakistan
2	Tehran University of Medical Sciences	31	5.92%	652	15	21.03	Iran
3	Pasteur Institute of Iran	25	4.77%	532	15	21.29	Iran
4	Universidade de São Paulo	23	4.39%	318	11	13.81	Brazil
5	Central Drug Research Institute India	21	4.01%	713	17	33.95	India
6	Universidade Federal de Minas Gerais	20	3.82%	488	11	24.4	Brazil
7	Kerman University of Medical Sciences	19	3.63%	437	10	23	Iran
8	Fundacao Oswaldo Cruz	19	3.63%	406	11	21.4	Brazil
9	Shiraz University of Medical Sciences	18	3.44%	276	9	15.33	Iran
10	Universidade Federal do Rio de Janeiro	17	3.24%	314	9	18.47	Brazil

The abbreviations included in Table 3 and Table 4, except for the h-index (as measured by the number of documents with at least ‘h’ citations).

**Table 6 diseases-11-00153-t006:** Top ten most productive journals for NPs for leishmaniasis treatment and research.

#	Journal	IF *	Citescore *	NP	Contribution Rate (%)	TC	h-Index	g-Index	m-Index
1	International Journal of Nanomedicine	7.033	10.9	21	4.01%	1163	15	21	1.07
2	Nanomedicine	6.096	8.3	19	3.63%	546	13	19	1.08
3	Acta Tropica	3.222	5.5	16	3.05%	407	11	16	1.0
4	Journal of Drug Targeting	5.016	8.8	11	2.10%	460	10	11	
5	Experimental Parasitology	2.132	3.9	10	1.91%	310	9	10	0.75
6	International Journal of Pharmaceutics	6.51	9.6	10	1.91%	421	8	10	0.28
7	Frontiers in Cellular and Infection Microbiology	6.073	5.9	9	1.72%	39	3	6	0.2
8	Plos Neglected Tropical Diseases	4.781	6.8	8	1.53%	150	6	8	0.45
9	Scientific Reports	4.997	6.9	8	1.53%	168	7	8	1.17
10	Journal of Drug Delivery Science and Technology	5.062	6.3	7	1.34%	35	3	5	0.5
10	Journal of Parasitic Diseases	-	2.0	7	1.34%	36	4	6	0.5

* Metrics 2021. Table 3 contains available abbreviations. h-index: the number of documents with at least ‘h’ citations. g-index: represents the number of articles that include at least ‘g2’ citations. Divide the h-index by the number of years since the author’s first publication to get the m-index.

**Table 7 diseases-11-00153-t007:** List of authors with ten or more publications in studies of NPs for the treatment and research of leishmaniasis.

Authors	Country	TP	TC	C/P	h-Index	g-Index	m-Index
Dube, Anuradha	India	16	540	33.75	14	16	0.824
Khamesipour, Ali	Iran	15	281	18.73	10	15	0.625
Rafati, Sima	Iran	14	388	27.71	12	14	0.857
Doroud, Delaram	Iran	12	375	31.25	11	12	0.786
Abamor, Emrah Şefik	Turkey	13	650	50.00	9	13	0.692
Nadhman, Akhtar	Pakistan	11	341	31.00	10	11	0.909
Shahnaz, G.	Pakistan	11	419	38.09	10	11	0.909
Yasinzai, Masoom M.	Pakistan	11	409	37.18	10	11	0.909
Shinwari, Zabta Khan	Pakistan	10	472	47.20	9	10	1.286
Abbasi, Banzeer Ahsan	Pakistan	10	406	40.60	8	10	1.6
Bağırova, Melahat	Azerbaijan	10	591	59.10	8	10	0.615
Iqbal, Javed	Pakistan	10	406	40.60	8	10	1.6
Sundar, Santhanam	India	10	235	23.50	8	10	0.5
Jafaari, Mahmoud Reza	Iran	10	173	17.30	7	10	0.583

Abbreviations are available in Table 3 and Table 6.

**Table 8 diseases-11-00153-t008:** Top ten most-cited documents in leishmaniasis research on NPs.

No.	Document (Authors/Year/Journal)	DOI	Year	GC	C/Y	LC	LC/GC Ratio (%)
1	Müller RH, 2001, Adv Drug Deliv Rev	10.1016/S0169-409X(00)00118-6	2001	1223	53.17	12	0.98
2	Schairer DO, 2012, Virulence	10.4161/viru.20328	2012	337	28.08	2	0.59
3	Matea CT, 2017, Int J Nanomed	10.2147/IJN.S138624	2017	326	46.57	1	0.31
4	Allahverdiyev AM, 2011, Future Microbiol	10.2217/fmb.11.78	2011	279	21.46	14	5.02
5	Tiuman TS, 2011, Int J Infect Dis	10.1016/j.ijid.2011.03.021	2011	245	18.85	18	7.35
6	Kayser O, 2003, Int J Pharm	10.1016/S0378-5173(02)00686-5	2003	181	8.62	11	6.08
7	Allahverdiyev AM, 2011, Int J Nanomedicine	10.2147/ijn.s23883	2011	164	12.62	55	33.54
8	Van De Ven H, 2012, J Control Release	10.1016/j.jconrel.2012.05.037	2012	124	10.33	17	13.71
9	Khalil AT, 2018, Artif Cells Nanomed Biotechnol	10.1080/21691401.2017.1345928	2018	118	19.67	8	6.78
10	Savoia D, 2015, J Infect Dev Ctries	10.3855/jidc.6833	2015	117	13.00	3	2.56

Abbreviations are available in Table 3 and Table 6, except C/Y (citations per year). GC: Global citation, LC: local citation.

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
