# Peer review of "Trends in Nanoparticles for Leishmania Treatment: A Bibliometric and Network Analysis"

_diseases, 2023, doi:10.3390/diseases11040153_

Round 1

Reviewer 1 Report

Comments and Suggestions for Authors

The article struggles to correlate the disease to the bibliometric data.  In a very few points, the article could translate the knowledge obtained to help the scientific community that is involved with Neglected Diseases.

 Also, regarding the text it self:

Lack of text fluency (hard to rred)

Unconnected sentences

 Lack of objectivity (long-winded)

Lack of accuracy ( terms like irrelevant document, respected journals are used without the definition, however it is in an article that presents metrics as the major focus.)

 Misuse of the sections introduction, materials and methods and results.  The article goals are listed in the methods section, some results are also presented in the methods.

Comments on the Quality of English Language

The texts should be reduced by half, especially in the introduction.

Author Response

Dear Editor

Thank you for giving us an opportunity to present a revised version of our work Manuscript ID: diseases-2619511. We have reviewed the manuscript carefully in order to address the questions of the referees. Below you can find the “answer to referees” and enclosed (via MDPI) the “draft revision” according to this reviewers’ suggestions. The authors sincerely thanks to the referee for their useful comments, which contribute to an improved presentation of the results.

Please see below our Response to your comments and reviewers Comment.

Sincerely,

On Behalf of co-authors

Karel Diéguez-Santana.

Universidad Regional Amazónica Ikiam, Parroquia Muyuna km 7 vía Alto Tena, 150150, Tena-Napo, Ecuador. K. Diéguez-Santana, email: karel.dieguez@ikiam.edu.ec

Wood Engineering Department, University of Bio-Bio. Concepcion. Chile

Reviewer1

Comments and Suggestions for Authors

The article struggles to correlate the disease to the bibliometric data.  In a very few points, the article could translate the knowledge obtained to help the scientific community that is involved with Neglected Diseases. Also, regarding the text it self:

Comment 1_R1: Lack of text fluency (hard to read)

Response 1_R1: Many thanks to the reviewer for the suggestions given. We improved the wording of some sections to make them easier to read. See the red underlined sections of the "draft revision" document.

Comment 2_R1: Unconnected sentences

Response 2_R1: Many thanks to the reviewer for the suggestions. We connect sentences better

Comment 3_R1: Lack of objectivity (long-winded)

Response 3_R1: Many thanks to the reviewer for the suggestions. We summarised several sections to objectify the results. This can be seen in the red underlined sections of the draft revision document.

Comment 4_R1: Lack of accuracy (terms like irrelevant document, respected journals are used without the definition, however it is in an article that presents metrics as the major focus.)

Response 4_R1: Many thanks to the reviewer for the suggestions. We define the suggested terms. See the red underlined Line 152-154 (irrelevant document) and  Line 186-187 (respected journals) in the "draft review" document.

Comment 5_R1: Misuse of the sections introduction, materials and methods and results.  The article goals are listed in the methods section, some results are also presented in the methods.

Response 5_R1: Many thanks to the reviewer for the suggestions. The article goals appear in the last paragraph of the introduction section, what appears in the methods section is a brief description of citation analysis and bibliometric techniques. In the case of the results that appeared in the methods, we incorporated that part into the first paragraph of the results section. See the red underlined, Line 185-190 in the "draft review" document.

Comment 6_R1: Comments on the Quality of English Language. The texts should be reduced by half, especially in the introduction.

Response 6_R1: Many thanks to the reviewer for the suggestions. We reduce the text of the introduction, see for example, Line 46-112 underlined in red from the "review draft" document.

Reviewer 2 Report

Comments and Suggestions for Authors

Dear Authors,

your manuscript gives a significant contribution to this field.

The title indicates the aim of the manuscript and the abstract is well written. It clearly indicates the work objective, methodology and result of the study.

The introduction is also well written.

The objectives of the study are of interest and are in line with the scope of the journal.

The methodology is clear.

The manuscript is well organized.

The conclusions are consistent with the evidence and arguments presented.

The reference is appropriate.

In my opinion, the manuscript could be accepted for publication in Disease.

Author Response

Dear Editor

Thank you for giving us an opportunity to present a revised version of our work Manuscript ID: diseases-2619511. We have reviewed the manuscript carefully in order to address the questions of the referees. Below you can find the “answer to referees” and enclosed (via MDPI) the “draft revision” according to this reviewers’ suggestions. The authors sincerely thanks to the referee for their useful comments, which contribute to an improved presentation of the results.

Please see below our Response to your comments and reviewers Comment.

Sincerely,

On Behalf of co-authors

Karel Diéguez-Santana.

Universidad Regional Amazónica Ikiam, Parroquia Muyuna km 7 vía Alto Tena, 150150, Tena-Napo, Ecuador. K. Diéguez-Santana, email: karel.dieguez@ikiam.edu.ec

Wood Engineering Department, University of Bio-Bio. Concepcion. Chile

Reviewer2

Dear Authors, your manuscript gives a significant contribution to this field. The title indicates the aim of the manuscript and the abstract is well written. It clearly indicates the work objective, methodology and result of the study.

  1. The introduction is also well written.
  2. The objectives of the study are of interest and are in line with the scope of the journal.
  3. The methodology is clear.
  4. The manuscript is well organized.
  5. The conclusions are consistent with the evidence and arguments presented.
  6. The reference is appropriate.
  7. In my opinion, the manuscript could be accepted for publication in Disease.

Response 1-7: Many thanks to the reviewer for the favorable comments.

Reviewer 3 Report

Comments and Suggestions for Authors

Comments on “Trends in nanoparticles for leishmania treatment: a bibliometric and network analysis”

This paper is suitable for publication after improvement. 

1) Nanoparticles have high surface energy or geometrical energy, which enable their wide applications in disease treatment. The article should emphasize this fact in this review article by the geometrical potential theory. 

2) In 2008, a new theory called as the cell fractal geometry was proposed to elucidate the nanoparticles’ effect on virus’ fatalness, see “ Fatalness of virus depends upon its cell fractal geometry” published in  CHAOS SOLITONS & FRACTALS , especially Fig. 2 of “Nano-scale drugs absorbed on the surface of a virus” in the article. 

3) The nanoparticles size and distribution on the leishmania treatment should be discussed, maybe the two-scale fractal dimensions can be used as an index . 

4) Additionally the thermal effect should be considered during the leishmania treatment, please refer to the article “thermal therapy for eye diseases” published in Thermal Science in 2020. 

Author Response

Dear Editor

Thank you for giving us an opportunity to present a revised version of our work Manuscript ID: diseases-2619511. We have reviewed the manuscript carefully in order to address the questions of the referees. Below you can find the “answer to referees” and enclosed (via MDPI) the “draft revision” according to this reviewers’ suggestions. The authors sincerely thanks to the referee for their useful comments, which contribute to an improved presentation of the results.

Please see below our Response to your comments and reviewers Comment.

Sincerely,

On Behalf of co-authors

Karel Diéguez-Santana.

Universidad Regional Amazónica Ikiam, Parroquia Muyuna km 7 vía Alto Tena, 150150, Tena-Napo, Ecuador. K. Diéguez-Santana, email: karel.dieguez@ikiam.edu.ec

Wood Engineering Department, University of Bio-Bio. Concepcion. Chile

Reviewer3

Comments and Suggestions for Authors

Comments on “Trends in nanoparticles for leishmania treatment: a bibliometric and network analysis”. This paper is suitable for publication after improvement. 

Comment 1_R3: Nanoparticles have high surface energy or geometrical energy, which enable their wide applications in disease treatment. The article should emphasize this fact in this review article by the geometrical potential theory. 

Response 1_R3: Many thanks to the reviewer for the suggestions given. We briefly discuss the geometric potential theory and how it can contribute to the treatment of leishmania with NPs. See the red underlined lines 562-570 in the "draft review" document.

Comment 2_R3: In 2008, a new theory called as the cell fractal geometry was proposed to elucidate the nanoparticles’ effect on virus’ fatalness, see “ Fatalness of virus depends upon its cell fractal geometry” published in  CHAOS SOLITONS & FRACTALS , especially Fig. 2 of “Nano-scale drugs absorbed on the surface of a virus” in the article. 

Response 2_R3: Many thanks to the reviewer for the suggestions given. We briefly discussed the cellular fractal geometry and its influence for the treatment of parasites with NPs. We consulted the suggested reference and it was cited in the paper. See the red underlined lines 562-570 in the "draft review" document.

Comment 3_R3: The nanoparticles size and distribution on the leishmania treatment should be discussed, maybe the two-scale fractal dimensions can be used as an index. 

Response 3_R3: Many thanks to the reviewer for the suggestions given. We discussed the size and distribution of nanoparticles in leishmania treatment. See the red underlined lines 570-575 in the "draft review" document.

Comment 4_R3: Additionally the thermal effect should be considered during the leishmania treatment, please refer to the article “thermal therapy for eye diseases” published in Thermal Science in 2020. 

Response 4_R3: Many thanks to the reviewer for the suggestions given. We considered the thermal effect and how it can influence during leishmania treatment. We consulted the suggested reference and it was cited in the paper. See the red underlined lines 572-575 in the "draft review" document.

Reviewer 4 Report

Comments and Suggestions for Authors

The manuscript entitled "Trends in nanoparticles for leishmania treatment: a biblio- 2 metric and network analysis" is quite interesting, as it provides an overview of the development of nanoparticles for the treatment of leishmaniasis. Below are some suggestions for work:

1) In the introduction, I suggest that the questions in lines 106 - 113 be placed in a single paragraph.

2) Discuss further why Iran, Brazil, India, and Pakistanthese are the main sources of research in the development of nanoparticles for the treatment of leishmaniasis. Try to correlate it with the epidemiology of the disease.

Author Response

Dear Editor

Thank you for giving us an opportunity to present a revised version of our work Manuscript ID: diseases-2619511. We have reviewed the manuscript carefully in order to address the questions of the referees. Below you can find the “answer to referees” and enclosed (via MDPI) the “draft revision” according to this reviewers’ suggestions. The authors sincerely thanks to the referee for their useful comments, which contribute to an improved presentation of the results.

Please see below our Response to your comments and reviewers Comment.

Sincerely,

On Behalf of co-authors

Karel Diéguez-Santana.

Universidad Regional Amazónica Ikiam, Parroquia Muyuna km 7 vía Alto Tena, 150150, Tena-Napo, Ecuador. K. Diéguez-Santana, email: karel.dieguez@ikiam.edu.ec

Wood Engineering Department, University of Bio-Bio. Concepcion. Chile

Reviewer4

The manuscript entitled "Trends in nanoparticles for leishmania treatment: a biblio- 2 metric and network analysis" is quite interesting, as it provides an overview of the development of nanoparticles for the treatment of leishmaniasis. Below are some suggestions for work:

Comment 1_R4: In the introduction, I suggest that the questions in lines 106 - 113 be placed in a single paragraph.

Response 1_R4: Many thanks to the reviewer for the suggestions given. We placed in a single paragraph and merged with another part of the document the questions in lines 106-113. See the red underlined lines 96-105, in the "draft revision" document.

Comment 2_R4: Discuss further why Iran, Brazil, India, and Pakistan these are the main sources of research in the development of nanoparticles for the treatment of leishmaniasis. Try to correlate it with the epidemiology of the disease.

Response 2_R4: Many thanks to the reviewer for the suggestions given. We explained why these countries are the main sources of research in the development of nanoparticles for the treatment of leishmaniasis (tropical countries that have the endemic disease and have relevant statistics of major cases reported worldwide). In addition, correlating with the epidemiology of the disease, we analysed the latest statistics of VL and CL cases, (WHO 2023). See Figure 2 and lines 228-233, in the "draft revision" document.
